# Pathogenic Delivery: The Biological Roles of Cryptococcal Extracellular Vesicles

**DOI:** 10.3390/pathogens9090754

**Published:** 2020-09-16

**Authors:** Haroldo C. de Oliveira, Rafael F. Castelli, Flavia C. G. Reis, Juliana Rizzo, Marcio L. Rodrigues

**Affiliations:** 1Instituto Carlos Chagas, Fundação Oswaldo Cruz (Fiocruz), Rua Prof. Algacyr Munhoz Mader, 3775 CIC Curitiba/PR, Curitiba 81350-010, Brasil; holiveira@aluno.fiocruz.br (H.C.d.O.); rcastelli@aluno.fiocruz.br (R.F.C.); flavia.reis@fiocruz.br (F.C.G.R.); 2Programa de Pós-Graduação em Biologia Parasitária, Instituto Oswaldo Cruz, Fiocruz, Avenida Brasil, 4.365, Pavilhão Arthur Neiva–Manguinhos, Rio de Janeiro 21040-360, Brasil; 3Centro de Desenvolvimento Tecnológico em Saúde (CDTS), Fundação Oswaldo Cruz (Fiocruz), Av. Brasil, 4036–Prédio Da Expansão–8˚ Andar–Sala 814, Rio De Janeiro 21040-361, Brasil; 4Unité Biologie des ARN des Pathogènes Fongiques, Département de Mycologie, Institut Pasteur, 25-28 Rue du Dr Roux, 75015 Paris, France; juliana.rizzo-balancin@pasteur.fr; 5Instituto de Microbiologia Paulo de Góes (IMPG), Universidade Federal do Rio de Janeiro, Cidade Universitária, Rio de Janeiro 21941-902, Brasil

**Keywords:** extracellular vesicles, biogenesis, *Cryptococcus* spp., extracellular vesicle analysis

## Abstract

Extracellular vesicles (EVs) are produced by all domains of life. In fungi, these structures were first described in *Cryptococcus neoformans* and, since then, they were characterized in several pathogenic and non-pathogenic fungal species. Cryptococcal EVs participate in the export of virulence factors that directly impact the *Cryptococcus*–host interaction. Our knowledge of the biogenesis and pathogenic roles of *Cryptococcus* EVs is still limited, but recent methodological and scientific advances have improved our understanding of how cryptococcal EVs participate in both physiological and pathogenic events. In this review, we will discuss the importance of cryptococcal EVs, including early historical studies suggesting their existence in *Cryptococcus*, their putative mechanisms of biogenesis, methods of isolation, and possible roles in the interaction with host cells.

## 1. Vesicular Export: A General System of Extracellular Delivery of Biological Structures

Extracellular vesicles (EVs) are vehicles exporting molecules from cells to the extracellular milieu and this kind of transport has been observed in organisms from all domains of life [1]. EVs are round-shaped, bilayered lipid membranes loaded with a diverse nature of molecular classes, including proteins [2], lipids [3,4], glycans [4,5], nucleic acids [6], and pigments [7,8].

In eukaryotes, EVs can be classified according to their mechanism of biogenesis. Apoptotic bodies are extracellular vesicles larger than 1 𝜇m that are released when the producing cells undergo apoptosis [9]. Microvesicles or ectosomes, which range from 100 nm to 1 𝜇m, are originated by shedding at the plasma membrane level [10]. Exosomes range from 30 to 200 nm and they result from the fusion of multivesicular bodies (MVBs) with the plasma membrane [11].

EV-like particles have been observed in fungi since the advent of high-resolution techniques of electron microscopy. In *Cryptococcus neoformans*, EV-like structures were first observed in 1973 using freeze-etching electron microscopy [12]. Two types of periplasmic, round-shaped structures ranging from 100 to 150 nm and 200 to 600 nm in diameter were at that time called paramural bodies [12,13]. These particles had properties that were all compatible with the presently known EVs, including dimensions, morphology, and bilayered membranes. In 2000, vesicle-like compartments containing glucosylceramide were detected at the cell wall of *C. neoformans* [14]. In 2007, EVs were isolated from *C. neoformans* culture supernatants [4]. Compositional and morphological studies showed that cryptococcal EVs corresponded to structures composed of several molecules participating in both fungal metabolism and pathogenicity [4,7]. After the seminal studies with *C. neoformans*, EVs were described in several fungal genera, including *Candida* [3,15], *Histoplasma* [3], *Saccharomyces* [3], *Sporothrix* [3,16], *Paracoccidioides* [17], *Malassezia* [18], *Alternaria* [19], *Rhizopus* [20], *Trichophytum* [21], *Pichia* [22], *Aspergillus* [23,24], *Fusarium* [25], *Trichoderma* [26], and *Exophiala* [27].

The knowledge of EVs in fungi is in frank expansion [28]. This review will focus on cryptococcal EVs, and we will discuss the main findings in this field, from the early description of EV-like particles in the 1970s to current times (Figure 1).

## 2. EV Biogenesis and Secretory Pathways in *Cryptococcus*

Several studies indicated that fungal EVs are produced and released under the coordination of multiple mechanisms. EVs may be formed at different cellular sites [29], with the possible participation of both post-Golgi conventional secretion and unconventional secretory pathways [30]. Several comprehensive reviews on the regulation of secretory pathways in eukaryotes are available in the literature [31,32,33], and the details of these processes are not in the scope of this manuscript.

The conventional secretory pathway in eukaryotes results in the fusion of post-Golgi vesicles with the plasma membrane, and subsequent release of luminal molecules to the extracellular milieu [34,35]. This general process requires to a large extent the participation of members of the *SEC* gene family, which regulates the traffic from the endoplasmic reticulum (ER) to the Golgi, and then to the cell surface [34,36]. The role of the *SEC6* gene in EV formation was evaluated in *Cryptococcus* [37]. The disruption of *SEC6* resulted in the negative detection of EVs in *C. neoformans.* However, the functional connection between *SEC* genes and EV formation has not been established in *Cryptococcus* [37].

Exosome formation requires the maturation of endosomes into MVBs [11]. The later compartments can be targeted to the cell surface, allowing fusion with the plasma membrane and consequent release of luminal MVB vesicles in the outer space [30,38]. MVB formation requires the functionality of the endosomal sorting complex required for transport (ESCRT). The ESCRT pathway is very complex, and its functionality demands a series of finely regulated events [39,40]. Briefly, it initiates with the phosphorylation of phosphatidylinositol 3-phosphate at the endosomal membrane by the phosphatidylinositol 3-kinase Vps34, resulting in the formation of the ESCRT-0 subcomplex. ESCRT-0 formation will consequently regulate the formation of the ESCRT-I, -II, and -III subcomplexes that will finally result in MVB formation and EV release [39,40].

In fungi, the importance of the ESCRT complex for EV formation was suggested in strains where distinct genes regulating ESCRT functions were disrupted. In *Candida albicans,* the deletion of several genes related to the ESCRT complex resulted in a significant decrease in EV production [41]. In *C. neoformans*, a mutant strain lacking expression of Vps27, a component of the ESCRT-0 subcomplex, manifested abnormal vesicle traffic and release, resulting in an accumulation of MVBs in the cytosol [42]. The deletion of other genes belonging to the ESCRT pathway led to significant defects in the delivery of virulence factors associated with EVs [42,43,44,45]. The cryptococcal mutant strains *vps27Δ* (ESCRT-0), *vps23Δ* (ESCRT-I), and *snf7Δ* (ESCRT-III) had attenuated virulence in a mice model of cryptococcosis [42,43,44]. Although the impact of gene deletion on EV production was studied only in the *vps27Δ* strain [42], the attenuation of virulence in ESCRT mutants suggests important connections between the unconventional secretory pathway and the pathogenesis of *Cryptococcus*.

Other regulators of unconventional secretion were linked to EV formation in *C. neoformans*. The Golgi reassembly and stacking protein (GRASP), for instance, regulates EV cargo and dimensions in *Cryptococcus* [46]. In *C. neoformans*, a *graspΔ* mutant strain produced EVs with dimensions that significantly differed from those produced by wild-type cells [46]. This strain also manifested attenuated virulence [47] and a different RNA composition [46]. Autophagy regulators, which also participate in the formation of EVs in other eukaryotes, also participate in the formation of cryptococcal EVs. An *atg7Δ* strain manifested hypovirulence [48] and EVs produced by this strain had a slightly different RNA composition, in comparison with wild-type cells [46]. Similarly, the flippase Apt1, which plays an essential role in membrane architecture and, consequently, in secretory mechanisms [49,50], was required for correct EV formation and virulence in *C. neoformans* [49]. Together, these results strongly suggest that EV formation, virulence, and unconventional secretion are connected in *C. neoformans*.

## 3. Cell Wall Passage

In fungi, exported particles and molecules have to overcome the cell wall to reach the outer environment [4,29,51]. EVs supposedly use three putative mechanisms to cross the cell wall. First, EV accumulation in the periplasmic space would create a turgor pressure shoving the vesicles to pass across the naturally existing pores of the cell wall. Second, EVs could catalyze their passage across the cell wall using glycan hydrolases, including β-glucosidases and endochitinases. Third, EVs could use pore channels to reach the outer environment, by getting deformed to adapt to the pore morphology, and moving out through cytoskeleton-dependent mechanisms [3,13,51,52].

In *C. neoformans,* EVs were found to be released collectively or individually [29], but the exact mechanism explaining how they cross the cell wall has not been characterized. Microscopic analyses demonstrated vesicles near to damaged areas in the cell wall, but a clear association between cell wall breakage and vesicle passage has not been established [29]. Indeed, intra-wall vesicles in apparently intact regions were found in *C. neoformans* [4]. Microscopic observations also revealed that melanization in *C. neoformans* is associated with the accumulation of vesicle-like structures in the periplasmic space [29,53]. During this process, a significant reduction in the porosity of the cell wall was observed, and vesicles were observed crossing the cell wall directly [29,53]. In summary, there has been no evidence so far that cryptococcal vesicles use pore channels to reach the extracellular space, reinforcing the hypotheses of pressure-induced release and/or vesicle-mediated cell wall hydrolysis. The latter hypothesis has been recently validated in bacteria. In *Bacillus subtilis,* EV formation was demonstrated to be a result of endolysins that degraded bacterial peptidoglycan and generated cell wall holes, which finally facilitated EV release [54].

Cell wall porosity can directly impact the efficacy of EV export through the fungal wall. Therefore, the composition of the cell wall might affect EV release. In this sense, *C. neoformans* mutants lacking each of the eight putative chitin synthase genes (*CHS1-8*) had their ability to produce EVs analyzed in a recent study. The *C. neoformans* mutants indeed manifested variable cell wall defects, but the analysis of EV production was puzzling, since the pattern of EV detection in the *chsΔ* mutants was highly variable [55]. For instance, it was initially predicted that disruption of *CHS3*, a gene encoding a class IV synthase mainly responsible for chitin synthesis in *C. neoformans*, would be more efficient in releasing vesicles, based on its previously suggested enhanced cell wall porosity [56]. However, this mutant was the one with the lowest efficacy in EV release. Other mutants (*chs4Δ* and *chs5Δ*) with no apparent cell wall alterations produced high amounts of EVs. Therefore, the differences observed in the EV analysis were not a consequence of altered cell wall porosity, although the possibility that the mutant strains simply had different abilities to produce EVs could not be ruled out. These results efficiently illustrate the need for a better understanding of how EVs traverse the fungal cell wall. In this sense, a recent study demonstrated that the vast majority of cryptococcal EVs are decorated with mannoproteins [57], suggesting that vesicle composition is directly affected by the presence of cell wall components. These results formed the basis for the proposal of a novel structural model of cryptococcal EVs, in which the outer vesicular layer is composed of the capsular polysaccharide glucuronoxylomannan (GXM), with the lipid bilayer carrying a fibrillar, protein coat enriched with mannoproteins [57].

## 4. Bioactive Components of Cryptococcal EVs

The first virulence-associated component characterized in cryptococcal EVs was GXM [4,7,58], the main component of the polysaccharide capsule [59,60,61]. It is now known that approximately 70% of cryptococcal EVs are coated with GXM [57]. In contrast to most of the microbial polysaccharides, GXM is synthesized intracellularly, in the Golgi [58]. In *C. neoformans*, disruption of the *SAV1* gene, which encodes a homolog of the Sec4/Rab8 subfamily GTPases that conservatively regulates exocytosis in yeast, resulted in an accumulation of vesicles loaded with GXM in the cytosol [58]. Additionally, the treatment of *C. neoformans* cells with brefeldin A, an inhibitor of the Golgi-derived transport, inhibited capsule formation [62]. Finally, deletion of the gene encoding GRASP resulted in aberrant Golgi morphology and reduced GXM secretion, with a negative impact on capsule size and attenuation of virulence in in vitro and in vivo models [47]. Together, these results point to the participation of the Golgi in GXM synthesis and export to the cell surface. The extracellular stage of GXM traffic, however, was not studied until cryptococcal EVs were first characterized. Since GXM is a major extracellular component in the *Cryptococcus* genus, the above-mentioned results implied the existence of mechanisms of trans-cell wall export.

The deletion of genes related to EV export through the ESCRT complex directly impacted the *Cryptococcus* capsule. Disruption of *VPS34*, *VPS27*, *HSE1*, *VPS23*, *VPS22*, *VPS25*, *VPS20*, and *SNF7* genes led to a significant decrease in capsule size [42,44,45,63]. These results could be related to the observation of EVs altered in size distribution and reduced capsule dimensions in the *C. neoformans vps27Δ* strain [42]. In this sense, capsular growth was correlated with EV detection. We observed that induction of capsule growth in vitro was accompanied by an increase in the detection of EVs carrying GXM [4]. Robertson et al. (2012) found that the treatment of *C. neoformans* cells with EDTA resulted in a remarkable reduction in EV detection, and a significant reduction in capsular diameter [64]. On the other hand, a *C. neoformans* mutant strain lacking a putative G1/S cyclin (Cln1) displayed an abnormal increase in capsule size, and a significantly increased production of EVs [65].

The content of cryptococcal EVs has been also linked to capsule formation. Deletion of the *C. gattii* encoding a putative scramblase (Aim25) resulted in an increased capsule size [66]. No differences in the amount of EVs were observed in WT and mutant strains. However, an enrichment of a population of larger EVs with a significantly increased GXM concentration was detected in the mutant. Interestingly, the acapsular strain *cap67Δ* was more efficient in incorporating GXM from EVs obtained from the *aim25Δ* strain than the WT strain [66]. The importance of membrane regulators on the proper EV formation and GXM export was also suggested in studies of the Apt1 flippase in *C. neoformans*. Mutant strains produced EVs with lower concentration of GXM and had smaller capsules in vivo [49,50]. More recently, it has been suggested that *ZIP3*, a cryptococcal regulator of manganese homeostasis, also participates in EV formation, as concluded from the observation of a higher concentration of GXM in culture supernatants of *zip3Δ* mutants and a high production of EVs, with an enrichment of an EV population with higher dimensions [67]. Together, these studies suggest the existence of connections between EV production and export of the most important capsule component of *Cryptococcus* spp.

Laccase, the enzyme catalyzing melanin synthesis, is another major virulence factor of *Cryptococcus* associated with EVs [4,7]. In vitro, *C. neoformans* EVs incubated with the melanin precursor L-3,4-dihydroxyphenylalanine (L-DOPA) became melanized [8]. It has been further proposed that melanin can be synthesized inside vesicles [68]. Melanin synthesis inside vesicles could protect *Cryptococcus* cells from the toxic compounds produced during melanin polymerization [68,69].

The *C. neoformans* mutant strains *vps34Δ*, *vps27Δ*, and *hseIΔ*, all showing functional defects in the ESCRT-0 complex, failed to export laccase to the cell wall [42], which might suggest an association between exosome formation and melanization in *Cryptococcus*. These results might be related to those observed with a *C. neoformans sec6Δ* mutant. Sec6 is a protein involved in the polarized fusion of exocytic vesicles with the plasma membrane, and its disruption in *Cryptococcus* resulted in an increased formation of MVB-like structures, affecting the transport of laccase to the cell wall [37]. A similar interpretation can apply to urease, another EV-linked virulence factor of cryptococci [7]. Interruption of the ESCRT pathway by disruption of the *VPS27* gene in *C. neoformans* resulted in reduced urease activity in vitro [42], and the same phenotype was also observed in the *C. neoformans sec6Δ* mutant [37]. These studies reinforce the notion that both conventional and unconventional secretory pathways participate in the release of cryptococcal EV-associated virulence molecules.

The diversity of molecules inside the *Cryptococcus* EVs is not restricted to virulence factors. Several RNA subclasses were described in cryptococcal EVs [6,66,70,71]. The first evidence of the presence of RNA in *Cryptococcus* EVs was provided by Nicola et al. (2009) using an RNA-selective nucleic acid dye to stain vesicular structures [71]. Different subclasses of RNA were further described in cryptococcal EVs, including, small nuclear RNA, ribosomal RNA, transfer RNA, microRNA, long noncoding RNA, and messenger RNA [6,70,72]. Recently, Liu et al. (2020) showed that Cin1, a multidomain adaptor protein that regulates cryptococcal growth, intracellular transport, and the production of several virulence factors [73], also plays an important role in regulating RNA export in *C. deneoformans* [70]. RNA export in *C. neoformans* EVs relies on the unconventional secretory pathway. Disruption of GRASP in *C. neoformans* leads to a significant change in the RNA cargo in EVs when compared to the WT strain [46]. Since disruption of GRASP also resulted in decreased GXM export, these results reinforce the notion that EVs and unconventional secretory mechanisms are connected in *Cryptococcus*. Figure 2 illustrates the importance of vesicles and EV cargo in physiology and virulence of *Cryptococcus*.

The participation of cryptococcal EVs and their components in fungal virulence suggests that targeting proteins participating in the secretory machinery could lead to the development of novel chemotherapies. Pharmacological inhibitors of EV formation in fungi have not been characterized so far. However, in other eukaryotes, compounds reported to inhibit EV formation (microvesicles or exosomes) were characterized [74]. If these molecules can also affect EV formation in *Cryptococcus* and other fungi, they could interfere with their pathogenic potential.

## 5. Impact of EVs during *Cryptococcus* Infection of Host Cells

EVs can interfere with the outcome of the interaction of cryptococci with infected cells. Murine macrophages RAW 264.7 and J774 can incorporate *C. neoformans* EVs [75,76]. Similarly, *C. gattii* EVs were incorporated by J774 macrophages [77]. The uptake of EVs by mouse macrophages is very efficient, as concluded from the incorporation of *C. gattii* EVs in only five minutes [77]. Actin polymerization inhibitors blocked EV uptake, suggesting the participation of cytoskeleton plasticity [77].

Exposure to cryptococcal EVs resulted in alterations of phagocyte functionality. The treatment of RAW 264.7 macrophages with *C. neoformans* EVs resulted in increased phagocytosis of non-opsonized *C. neoformans* [75]. A more prominent increase in the phagocytosis levels was observed when the macrophages were stimulated with EVs produced by a *C. neoformans* acapsular strain, which indicates that changes in vesicular composition differentially impact their functions [75]. In the same study, *C. neoformans* EVs were demonstrated to affect cytokine production by RAW 264.7 macrophages. Stimulation of the macrophages with EVs led to increased production of TNF-α, TGF-β, and IL-10. Once again, differences were found between stimulation of the phagocytes with EVs from acapsular or encapsulated *C. neoformans* strains. EVs from the acapsular strain led to an increase in the production of TNF-α, which induced antifungal activity. On the other hand, EVs from the encapsulated *C. neoformans* strain led to a significant increase in the production of TGF-β and IL-10, which are known to be positively modulated by GXM [75]. *C. neoformans* EVs also modulated nitric oxide (NO) production. Curiously, the stimulation of NO production was significantly less effective when the macrophages were treated with EVs isolated from the acapsular *C. neoformans* strain *cap67Δ* [75]. These results might be related to the ability of the EVs to modulate fungal killing by host phagocytes. Accordingly, environmental phagocytes are also affected by *Cryptococcus* EVs. Rizzo et al. (2017) observed stimulation of *Acanthamoeba castellanii* with EVs resulted in a significantly increased survival of phagocytized *C. neoformans* [78].

Besides influencing the performance of phagocytes, cryptococcal EVs also modulated important features of the *Cryptococcus* physiology during macrophage infection. *C. gattii* EVs obtained from a virulent strain were used to treat macrophages infected with a non-virulent *C. gattii* isolate, which resulted in the accumulation of the vesicles in the phagosomes [77]. Inside the phagosomes, the EVs from the pathogenic *C. gattii* strain stimulated the intracellular replication of the non-pathogenic isolate. Negative results were observed when EVs from the non-pathogenic strain or produced by an acapsular mutant were tested [77]. Similarly, Hai et al. (2020) demonstrated that culture filtrates from a high virulent strain induced an increased virulence in a hypovirulent strain [79]. This effect was only observed under conditions of EV availability. These results indicate that cryptococcal EVs are vehicles operating in the transfer of virulence traits between distinct *Cryptococcus* strains and demonstrate an important function of the vesicles in cell-to-cell communication processes.

*Cryptococcus* EVs were also suggested to positively impact both adhesion and invasion of the blood–brain barrier (BBB) by fungal cells [80]. In a mice model, *C. neoformans* EVs induced an enhanced fungal burden in the brain and the cerebrospinal fluid in a dose-dependent manner, with an accumulation of structures that could correspond to EVs surrounding the brain lesions on infected mice [80]. More recently, additional modulatory effects on the host’s immune mechanisms were demonstrated. The mammalian β-galactoside-binding protein Galectin-3 (Gal-3) recognized EVs and promoted vesicle disruption, resulting in decreased levels of interaction of the fungi with macrophages in vitro, reduced recovery of intact EVs, and a diminished uptake of EVs by macrophages [81].

## 6. Cryptococcal EVs: Vaccine Candidates?

The ability of cryptococcal vesicles to modulate the host’s immunological functions might result in vaccinal potential. Since licensed antifungal vaccines are still not available [82], information on how fungal EVs activate the immune response could be greatly impactful. The vaccinal potential of fungal EVs was first suggested in the *Candida* model [83,84], and similar observations were recently described in *Cryptococcus*. In an immunization model of *Galleria mellonella* with vesicular structures enriched in sterolglycosides (SGs) and GXM, EV administration resulted in the protection of the invertebrate host against a lethal challenge with *C. neoformans* [85], revealing a potential vaccination strategy for cryptococcosis using *sgl1∆* EVs [85]. The vaccinal potential of cryptococcal EVs was recently confirmed in a murine model of cryptococcosis. Mice immunized with EVs obtained from an acapsular *C. neoformans* mutant strain induced a strong antibody response and significantly prolonged survival of mice upon a lethal challenge with *C. neoformans* [57]. Importantly, the immunological mechanisms associated with this protection are still unknown, but cryptococcal EVs were recognized by antibodies produced by infected mice [7,57]. Figure 3 presents an overview of the role of EVs during the interaction of *Cryptococcus* with host cells, including their vaccinal potential.

## 7. Facilitated Methods for the Analysis of Cryptococcal Vesicles

The generation of knowledge on the functions and mechanisms of the biogenesis of cryptococcal vesicles has been continuously affected by methodological limitations. Empirically, it is known in the field that *Cryptococcus* EVs are produced in low yields, in comparison to other models. Indeed, our laboratory experience shows that other yeast genera, including *Saccharomyces* and *Candida*, are more efficient producers of EVs. Therefore, the perception that improved methods of EV analysis were necessary for the *Cryptococcus* model has been clear for years.

EV analysis in fungi and other eukaryotes has historically included isolation of membrane structures from the supernatants of liquid cultures by ultracentrifugation methods, followed by particle analysis by a combination of microscopic and physical methods [4]. Fungal EVs have been analyzed according to these protocols for more than a decade. Although this model has been helpful to address several questions, it must be highlighted that fungal cells are rarely distributed in liquid matrices both in the environment and during infection. The isolation of cryptococcal EVs from liquid media can take up to two weeks, with very low yields of EV isolation. This was the basis for the design of a novel protocol of isolation of EVs from the *Cryptococcus* genus. We hypothesized that EVs could be recovered from cultures obtained in solid media since there was no evidence in the literature suggesting that EVs were exclusively produced in liquid matrices.

Cultivation of *C. neoformans* and *C. gattii* in regular agar plates followed by suspension of yeast cells in PBS for further centrifugation steps resulted in facilitated detection of typical EVs [66]. However, since this study and earlier articles used diverse methods for EV quantification, a reliable comparison between the yields of the different methods is still not available. Of note, the solid medium protocol successfully allowed EV detection independently on the medium used, and all fungal species tested, including *Candida albicans*, *Histoplasma capsulatum*, and *Saccharomyces cerevisiae*, gave positive results. The protocol was shown to be highly reproducible, and fast: from the recovery of fungal cells to the analysis of ultracentrifugation pellets, the time estimated was of 5 h. Isolated EVs were reliably detected by ELISA targeting GXM, nanoparticle tracking analysis, and transmission electron microscopy. Our most recent unpublished results indicate that the facilitated EV isolation method allows efficient analysis of samples obtained from multiple isolates, separation of vesicles by gradient centrifugation, and a small molecule composition. We anticipate that, in this new scenario, it will be possible to experimentally address currently complicated questions related to vesicle fractionation, diversity, and biogenesis. The most recent methods of EV isolation from cryptococcal cultures are summarized in Figure 4.

## 8. Gaps, Unanswered Questions, and Perspectives

Despite the recent progress in the field of fungal EVs, in particularly in the *Cryptococcus* model, it is unquestionable that several gaps and questions remain open. For instance, most studies performed so far were based on single, standard strains of *C. neoformans* rather than *C. gattii*, which limits our knowledge on the compositional diversity of cryptococcal EVs. Considering that EV composition is a major determinant of their functions, studies on the diversity in the production of EVs by different cryptococcal species and strains are necessary. Similarly, it is still unknown whether the production of EVs changes at the various life-stages of *Cryptococcus*.

Novel methods for EV separation are similarly necessary. All studies performed with *Cryptococcus* so far used centrifugation protocols that resulted in the coisolation of diverse EV populations, as recently illustrated in early [7] and recent [57] studies. This limitation directly impacts, for instance, immunological studies, since these studies are testing mixed EV populations that can manifest divergent immunological functions. Therefore, methods separating EVs based on their biogenesis and/or physical chemical properties are required for refining the functional studies, and they likely improve the knowledge on their vaccinal potential.

Finally, as previously mentioned in this manuscript and several others, we do know that cryptococcal EVs have different cellular origins, but we still do not know where exactly they come from. The identification of genes regulating EV formation and/or pharmacological inhibitors of EV release in *Cryptococcus* will likely open new venues of investigation, with the potential to change the way we understand the functions of cryptococcal EVs.

## Figures and Tables

**Figure 1 pathogens-09-00754-f001:**
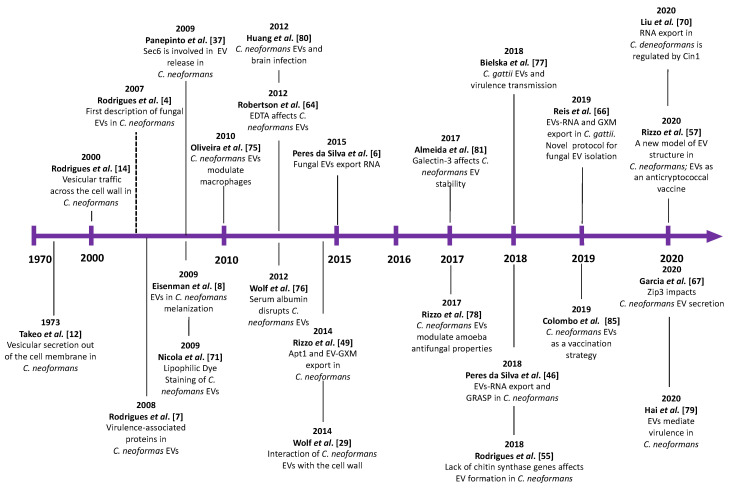
Main discoveries related to cryptococcal extracellular vesicles (EVs). The timeline herein illustrated for *Cryptococcus* findings was based on that described for all fungal EVs [28].

**Figure 2 pathogens-09-00754-f002:**
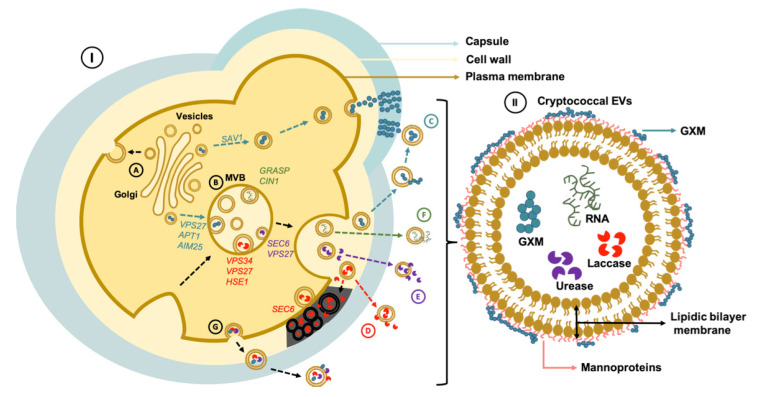
EV export and transport of virulence traits in *Cryptococcus* spp. (I) Vesicle-mediated export of cryptococcal virulence factors are impacted by both conventional (A) and unconventional (B) secretory pathways. (C) Transport of GXM, a major virulence factor that is transported in vesicles, is impacted by both pathways, as illustrated by the need of the *SAV1* gene (conventional secretion), and the possible involvement of the *VPS27* gene (unconventional secretion) for polysaccharide export. GXM-containing EVs in the outer space might operate as a polysaccharide source for capsule construction. At least two proteins related to membrane architecture and EV formation (Aim25 and Apt1) participate in GXM export. (D) Laccase export is affected by the expression of genes operating in the ESCRT complex (*VPS34*, *VPS27*, and *HSEI*). *SEC6*, a component of the secretory pathway, also participate in vesicle-mediated melanization. (E) Urease is another virulence factor exported in EVs and its activity is also influenced by genes participating in conventional (*SEC6*) and unconventional (*VPS27*) secretory pathways. (F) RNA is also exported in EVs (H) through mechanisms that require *CIN1* and *GRASP*. (G) EVs carrying virulence factors of *Cryptococcus* may also originate directly from the plasma membrane. However, the molecular mechanisms behind this process are still unknown. (II) Illustration of a cryptococcal EV.

**Figure 3 pathogens-09-00754-f003:**
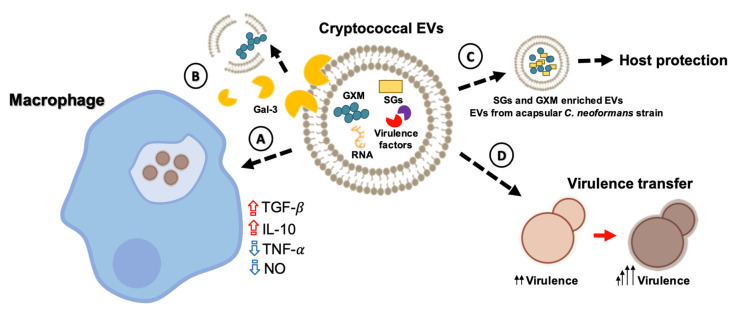
*Cryptococcus* EVs are biologically active and modulate the activity of immune cells. Cryptococcal EVs containing diverse compounds leave the fungal cell and are incorporated by host phagocytes, where they have a number of modulatory activities. (A) EVs can reach the intracellular space, where their cargo modulates the host response, as illustrated by the increased production of TGF-β and IL-10, and decreased release of TNF-α and nitric oxide (NO). A possible outcome of EV exposure is a defective antifungal response by the macrophages, favoring cryptococcal intracellular replication. (B) Galectin-3 (Gal-3) can interact with EVs and disrupt them, leading to failure in the delivery of concentrated virulence factors into host cells and tissues. (C) Cryptococcal EVs represent a potential vaccine strategy to cryptococcosis. EVs from an acapsular *C. neoformans* mutant or enriched with sterylglucosides (SGs) favor the survival of mice and *G. mellonella* after a lethal challenge with *C. neoformans*, respectively. (D) *Cryptococcus* EVs also play a role in cell-to-cell communication, turning hypovirulent *C. gattii* cells into a virulent population.

**Figure 4 pathogens-09-00754-f004:**
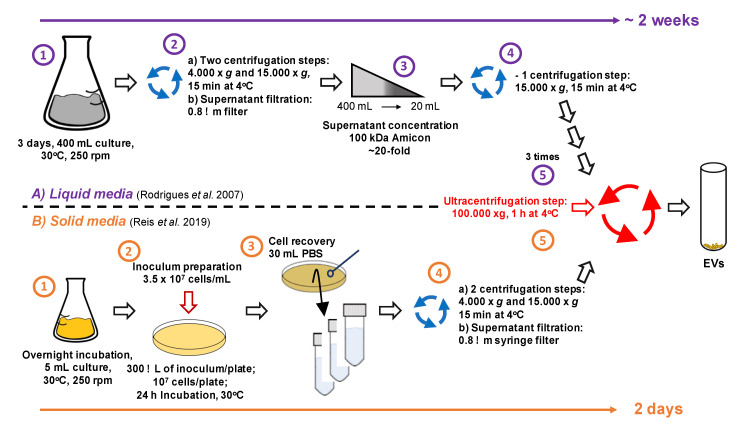
Protocols to isolate cryptococcal EVs. The upper panel (**A**) summarizes EV isolation from liquid cultures. The lower panel (**B**) summarizes the protocol used for EV isolation from agar plates. For details, please visit the references mentioned in panel (**A**) [4] and (**B**) [66].

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
