# Peer review of "Pathogenic Delivery: The Biological Roles of Cryptococcal Extracellular Vesicles"

_pathogens, 2020, doi:10.3390/pathogens9090754_

Round 1

Reviewer 1 Report

The manuscript by de Oliveira and Castelli et al reviews the current knowledge of extracellular vesicles (EVs) from the fungus Cryptococcus neoformans. It is a comprehensive review which contain all recent findings and very nice and clear drawings. Unfortunately, there are lots of wrong citations in the text and they need to be corrected. Please check again all the citations. Below you will find my suggestions.

Lines 29, 30: All living organisms can deliver proteins, glycans, and many other classes of molecules to the extracellular space [1]. Citation [1] does not review all living organisms but only Gram-positive bacteria, mycobacteria and fungi. Additionally, this sentence, contains ‘and many other classes of molecules’ which is very vague and should be avoided.
My suggestion is to delete this very first sentence.

Lines 30, 31: Extracellular vesicles (EVs) are important vehicles participating in the export of molecules to the extracellular milieu [1,2]. Citation 2. should be removed here, but then citation 1, again, does not include all domains of life.

Lines 31, 32: EV-mediated transport is conserved across all domains of life [1,3–7].

1. Brown, L.; Wolf, J.M.; Prados-Rosales, R.; Casadevall, A. Through the wall: extracellular vesicles in Gram- positive bacteria, mycobacteria and fungi. Nat. Rev. Microbiol 2015, 13, 620–630, doi:10.1038/nrmicro3480.

3. Gill, S.; Catchpole, R.; Forterre, P. Extracellular membrane vesicles in the three domains of life and beyond. FEMS Microbiol Rev. 2019, 43, 273–303, doi:10.1093/femsre/fuy042.

4. Rodrigues, M.L.; Godinho, R.M.; Zamith-Miranda, D.; Nimrichter, L. Traveling into Outer Space: Unanswered Questions about Fungal Extracellular Vesicles. PLoS Pathog 2015, 11, e1005240, doi:10.1371/journal.ppat.1005240.

5. Eitan, E.; Suire, C.; Zhang, S.; Mattson, M.P. Impact of lysosome status on extracellular vesicle content and release. Ageing Res. Rev. 2016, 32, 65–74, doi:10.1016/j.arr.2016.05.001.

6. Rodrigues, M.L.; Casadevall, A. A two-way road: novel roles for fungal extracellular vesicles. Mol. Microbiol 2018, 110, 11–15, doi:10.1111/mmi.14095.

7. Yoon, Y.J.; Kim, O.Y.; Gho, Y.S. Extracellular vesicles as emerging intercellular communicasomes. BMB Rep. 2014, 47, 531–539, doi:10.5483/bmbrep.2014.47.10.164.

I would replace the citations 4, 5 (especially 5!), 6, and 7 (as they are about novel roles of EVs or EV-communication) with Woith E, Fuhrmann G, Melzig MF. Extracellular Vesicles-Connecting Kingdoms. Int J Mol Sci. 2019;20(22):5695. Published 2019 Nov 14. doi:10.3390/ijms20225695.

Second and third sentences could be combined to one sentence e.g. Extracellular vesicles (EVs) are vehicles exporting molecules from cells to the extracellular milieu and this kind of unconventional transport has been observed in organisms from all domains of life [Woith E, Fuhrmann G, Melzig MF. Extracellular Vesicles-Connecting Kingdoms. Int J Mol Sci. 2019;20(22):5695. Published 2019 Nov 14. doi:10.3390/ijms20225695]

Lines 32, 33: EVs are round-shaped, bilayered lipid membranes loaded with a diverse nature of molecular classes, including proteins, lipids, glycans, nucleic acids, and pigments [7–11]. None of the citations 7-11 is correct.

Lines 33, 37: The existence of EVs was suggested in early studies with blood cells. Wolf, in 1967, described vesicle-like structures that were at that time called “platelet dust” after centrifugation of human plasma [12,13]. Similar structures were observed in tissues [14], body fluids [15], and other types of mammalian cells [10]. I do not think this sentence is important here at all (as the main subject are cryptococcal EVs), and again, several citations here are not correct.

I would strongly suggest removing text describing non-fungal EVs from the introduction (lines 34-55). It does not improve the manuscript significantly and again, there are problems with the citations.

Starting with the line 56 after line the 33 will work very well.

Line 60: These particles had properties that were all compatible with the presently known EVs. Please describe the properties here.

Lines 62-64: Compositional and morphological studies showed that cryptococcal EVs corresponded to bag-like structures composed of several molecules participating in both fungal metabolism and pathogenicity [2,24]. Compositional study cannot correspond to bag-like structures. Please rephrase this sentence.

Lines 79, 80: Several comprehensive reviews on the regulation of secretory pathways in eukaryotes are available in the literature [41–45], and the details of these processes are not in the scope of this manuscript. In the publications 41-45 we have mainly yeasts, including C. neoformans, so it is definitely a scope of this manuscript:

41. Delic, M.; Valli, M.; Graf, A.B.; Pfeffer, M.; Mattanovich, D.; Gasser, B. The secretory pathway: exploring yeast diversity. FEMS Microbiol. Rev. 2013, 37, 872–914, doi:10.1111/1574-6976.12020.

42. Shikano, S.; Colley, K.J. Secretory Pathway. In Encyclopedia of Biological Chemistry; Elsevier, 2013; pp. 203– 209.

43. Rodrigues, M.L.; Djordjevic, J.T. Unravelling Secretion in Cryptococcus neoformans: More than One Way to Skin a Cat. Mycopathologia 2012, 173, 407–418, doi:10.1007/s11046-011-9468-9.

44. Rollenhagen, C.; Mamtani, S.; Ma, D.; Dixit, R.; Eszterhas, S.; Lee, S.A. The Role of Secretory Pathways in Candida albicans Pathogenesis. J. Fungi 2020, 6, 26, doi:10.3390/jof6010026.

45. Novick, P.; Ferro, S.; Schekman, R. Order of events in the yeast secretory pathway. Cell 1981, 25, 461–469, doi:10.1016/0092-8674(81)90064-7.

Lines 81-83: The conventional secretory pathway in eukaryotes results in the fusion of post-Golgi vesicles with the plasma membrane, and subsequent release of luminal molecules to the extracellular milieu [40,46]. Wrong citations.

Lines 88-89: The unconventional secretory pathway includes several mechanisms of extracellular protein release, many of them involving EVs [40,46]. Wrong citations. Again, this is a very elusive piece of information and should be more precise.

Lines 89-90: Exosome formation, one of the most studied mechanisms of unconventional secretion … Exosome formation is not a mechanism of unconventional secretion, this is just exosome formation. Please rephrase.

Lines 111-112: (…) regulates EV formation in many eukaryotes [40,46,57,58]. The citations 40,46,57,58 represent only fungal species, so I suggest to rephrase the text or to change citations.

Lines starting from 144 about chitin synthases – could you please mention very briefly the function of each chs mentioned in the text (can be in brackets), this will provide some hints for the reader about potential implications on EV formation in chs respective mutants.

Line 150: ‘gave high signals of EV detection’ – not sure what that means.

Lines 163-168: In contrast to most of the microbial polysaccharides, GXM is synthesized intracellularly, in the Golgi [67]. In C. neoformans, disruption of the SAV1 gene, which encodes a homolog of the Sec4/Rab8 subfamily GTPases that conservatively regulates exocytosis in yeast, resulted in an accumulation of vesicles loaded with GXM in the cytosol [67]. Also, the treatment of C. neoformans cells with brefeldin A, an inhibitor of the Golgi-derived transport, inhibited capsule formation [71]. I think these lines could be transferred to the biogenesis part.

…. There is lots of overlapping between Bioactive components of cryptococcal EVs and EV Biogenesis and Secretory Pathways in Cryptococcus. I am keen to combine them together in one chapter, but I will leave this decision to the Authors.

Line 247: Besides their influence on Cryptococcus physiology, EVs …. What exactly did authors mean here?

Line 326: This particularity in addition to the fact that isolation of cryptococcal EVs from liquid media can take up to two weeks to produce very low yields of vesicle isolation consisted of the experimental basis for the design of a novel protocol of isolation of EVs from the Cryptococcus genus. This sentence is too long.

Lines 333-334: Cultivation of C. neoformans and C. gattii in regular agar plates followed by suspension of yeast cells in PBS for further centrifugation steps resulted in high-yield detection of typical EVs [75]. Could you please add the information (if known) how much the two methods differ in EVs yield obtained at the end of each isolation process?

Figure 3: I have few suggestions here:

1) Could you please add the localisation of GXM and mannoproteins to the EV picture (II)? (based on [66]).
2) Could the fonts in the figure (I) be bigger, please?
3) (B), an unconventional secretory pathway is represented by an arrow directed towards the cell, which suggests an uptake, not secretion. I am not sure how to change it, maybe by removing the arrow and putting B letter next to MVB or next to the arrow coming from the MVB?
4) It is uncommon that EVs have other shapes than round (based on [66]), so it is extremely important to show it here. I suggest to draw few different shapes according to [66].

Figure 4: The description of the vesicle C is not clear; there is also a typo in EVs (it is written as Evs)

Figure 5: misses μ in μm 

References: Please italicize species names.

Please decide if in vitro and in vivo is written in italics or not throughout the text.

Author Response

Reviewer 1 Comment: The manuscript by de Oliveira and Castelli et al reviews the current knowledge of extracellular vesicles (EVs) from the fungus Cryptococcus neoformans. It is a comprehensive review which contain all recent findings and very nice and clear drawings. Unfortunately, there are lots of wrong citations in the text and they need to be corrected. Please check again all the citations. Below you will find my suggestions.

Authors' Response: Thank you for pointing this out. We apologize for this major mistake and corrected all citations, as suggested by the reviewer.

Reviewer 1 Comment: Lines 29, 30: All living organisms can deliver proteins, glycans, and many other classes of molecules to the extracellular space [1]. Citation [1] does not review all living organisms but only Gram-positive bacteria, mycobacteria and fungi. Additionally, this sentence, contains ‘and many other classes of molecules’ which is very vague and should be avoided. My suggestion is to delete this very first sentence.

Authors' Response: We agree with the reviewer and removed the first sentence. 

Reviewer 1 Comment: Lines 30,31: Extracellular vesicles (EVs) are important vehicles participating in the export of molecules to the extracellular milieu [1,2]. Citation 2. should be removed here, but then citation 1, again, does not include all domains of life.

Lines 31, 32: EV-mediated transport is conserved across all domains of life [1,3–7].

1. Brown, L.; Wolf, J.M.; Prados-Rosales, R.; Casadevall, A. Through the wall: extracellular vesicles in Gram- positive bacteria, mycobacteria and fungi. Nat. Rev. Microbiol 2015, 13, 620–630, doi:10.1038/nrmicro3480.

3. Gill, S.; Catchpole, R.; Forterre, P. Extracellular membrane vesicles in the three domains of life and beyond. FEMS Microbiol Rev. 2019, 43, 273–303, doi:10.1093/femsre/fuy042.

4. Rodrigues, M.L.; Godinho, R.M.; Zamith-Miranda, D.; Nimrichter, L. Traveling into Outer Space: Unanswered Questions about Fungal Extracellular Vesicles. PLoS Pathog 2015, 11, e1005240, doi:10.1371/journal.ppat.1005240.

5. Eitan, E.; Suire, C.; Zhang, S.; Mattson, M.P. Impact of lysosome status on extracellular vesicle content and release. Ageing Res. Rev. 2016, 32, 65–74, doi:10.1016/j.arr.2016.05.001.

6. Rodrigues, M.L.; Casadevall, A. A two-way road: novel roles for fungal extracellular vesicles. Mol. Microbiol 2018, 110, 11–15, doi:10.1111/mmi.14095.

7. Yoon, Y.J.; Kim, O.Y.; Gho, Y.S. Extracellular vesicles as emerging intercellular communicasomes. BMB Rep. 2014, 47, 531–539, doi:10.5483/bmbrep.2014.47.10.164.

I would replace the citations 4, 5 (especially 5!), 6, and 7 (as they are about novel roles of EVs or EV-communication) with Woith E, Fuhrmann G, Melzig MF. Extracellular Vesicles-Connecting Kingdoms. Int J Mol Sci. 2019;20(22):5695. Published 2019 Nov 14. doi:10.3390/ijms20225695.

Authors' Response: As suggested by the reviewer, we modified the manuscript and replaced the references. Please see topic 1 of the revised manuscript.

Reviewer 1 Comment: Second and third sentences could be combined to one sentence e.g. Extracellular vesicles (EVs) are vehicles exporting molecules from cells to the extracellular milieu and this kind of unconventional transport has been observed in organisms from all domains of life.

Authors' Response: Done, as requested by the reviewer.

Reviewer 1 Comment: Lines 32, 33: EVs are round-shaped, bilayered lipid membranes loaded with a diverse nature of molecular classes, including proteins, lipids, glycans, nucleic acids, and pigments [7–11]. None of the citations 7-11 is correct.

Authors' Response: We fully revised the citations, following the reviewer's recommendation. 

Reviewer 1 Comment: Lines 33, 37: The existence of EVs was suggested in early studies with blood cells. Wolf, in 1967, described vesicle-like structures that were at that time called “platelet dust” after centrifugation of human plasma [12,13]. Similar structures were observed in tissues [14], body fluids [15], and other types of mammalian cells [10]. I do not think this sentence is important here at all (as the main subject are cryptococcal EVs), and again, several citations here are not correct.

I would strongly suggest removing text describing non-fungal EVs from the introduction (lines 34-55). It does not improve the manuscript significantly and again, there are problems with the citations. Starting with the line 56 after line the 33 will work very well.

Authors' Response: We removed the "non-fungal text", as suggested by the reviewer.

Reviewer 1 Comment: Line 60: These particles had properties that were all compatible with the presently known EVs. Please describe the properties here.

Authors' Response: Done, please see lines 43-433.

Reviewer 1 Comment: Lines 62-64: Compositional and morphological studies showed that cryptococcal EVs corresponded to bag-like structures composed of several molecules participating in both fungal metabolism and pathogenicity [2,24]. Compositional study cannot correspond to bag-like structures. Please rephrase this sentence.

Authors' Response: Done, please see lines 44-46.

Reviewer 1 Comment: Lines 79, 80: Several comprehensive reviews on the regulation of secretory pathways in eukaryotes are available in the literature [41–45], and the details of these processes are not in the scope of this manuscript. In the publications 41-45 we have mainly yeasts, including C. neoformans, so it is definitely a scope of this manuscript.

Authors' Response: We fully revised the citations, following the reviewer's recommendation. 

Reviewer 1 Comment: Lines 81-83: The conventional secretory pathway in eukaryotes results in the fusion of post- Golgi vesicles with the plasma membrane, and subsequent release of luminal molecules to the extracellular milieu [40,46]. Wrong citations.

Authors' Response: We fully revised the citations, following the reviewer's recommendation

Reviewer 1 Comment: Lines 88-89: The unconventional secretory pathway includes several mechanisms of extracellular protein release, many of them involving EVs [40,46]. Wrong citations. Again, this is a very elusive piece of information and should be more precise.

Authors' Response: We fully revised the citations, following the reviewer's recommendation and modified this sentence for clarity.

Reviewer 1 Comment: Lines 89-90: Exosome formation, one of the most studied mechanisms of unconventional secretion ... Exosome formation is not a mechanism of unconventional secretion, this is just exosome formation. Please rephrase.

Authors' Response: Rephrased as required.

Reviewer 1 Comment: Lines 11-112: (...) regulates EV formation in many eukaryotes [40,46,57,58]. The citations 40,46,57,58 represent only fungal species, so I suggest to rephrase the text or to change citations.

Authors' Response: Rephrased as required.

Reviewer 1 Comment: Lines starting from 144 about chitin synthases – could you please mention very briefly the function of each chs mentioned in the text (can be in brackets), this will provide some hints for the reader about potential implications on EV formation in chs respective mutants.

Authors' response: Thank you for pointing this out. Please see 125-133 of the revised manuscript.

Reviewer 1 Comment: Lines 163-168: In contrast to most of the microbial polysaccharides, GXM is synthesized intracellularly, in the Golgi [67]. In C. neoformans, disruption of the SAV1 gene, which encodes a homolog of the Sec4/Rab8 subfamily GTPases that conservatively regulates exocytosis in yeast, resulted in an accumulation of vesicles loaded with GXM in the cytosol [67]. Also, the treatment of C. neoformans cells with brefeldin A, an inhibitor of the Golgi-derived transport, inhibited capsule formation [71]. I think these lines could be transferred to the biogenesis part.

Authors' Response: We tried to follow the reviewer's suggestion, but the text read really fragmented. We then appreciate this comment, but we opted for leaving GXM synthesis in this topic.

Reviewer 1 Comment: .... There is lots of overlapping between Bioactive components of cryptococcal EVs and EV Biogenesis and Secretory Pathways in Cryptococcus. I am keen to combine them together in one chapter, but I will leave this decision to the Authors.

Authors' Response: Following the other reviewers' comments, we believe this point was properly addressed - please see the highlighted text in the revised manuscript.

Reviewer 1 Comment: Line 247: Besides their influence on Cryptococcus physiology, EVs .... What exactly did authors mean here?

Authors' Response: This sentence was rephrased for clarity.

Reviewer 1 Comment: Line 326: This particularity in addition to the fact that isolation of cryptococcal EVs from liquid media can take up to two weeks to produce very low yields of vesicle isolation consisted of the experimental basis for the design of a novel protocol of isolation of EVs from the Cryptococcus genus. This sentence is too long.

Authors' Response: Rephrased as required.

Reviewer 1 Comment: Lines 333-334: Cultivation of C. neoformans and C. gattii in regular agar plates followed by suspension of yeast cells in PBS for further centrifugation steps resulted in high-yield detection of typical EVs [75]. Could you please add the information (if known) how much the two methods differ in EVs yield obtained at the end of each isolation process?

Authors' Response: This is an interesting point which is hard to address, but we modified the text to meet the reviewer's suggestion. Please see lines 333-335 of the revised manuscript. 

Reviewer 1 Comment: Figure 3: I have few suggestions here:

1) Could you please add the localisation of GXM and mannoproteins to the EV picture (II)? (based on [66]).
2) Could the fonts in the figure (I) be bigger, please?
3) (B), an unconventional secretory pathway is represented by an arrow directed towards the cell, which suggests an uptake, not secretion. I am not sure how to change it, maybe by removing the arrow and putting B letter next to MVB or next to the arrow coming from the MVB?
4) It is uncommon that EVs have other shapes than round (based on [66]), so it is extremely important to show it here. I suggest to draw few different shapes according to [66].

Authors' Response: Suggestions were fully incorporated into the revised figure.

Reviewer 1 Comment: Figure 4: The description of the vesicle C is not clear; there is also a typo in EVs (it is written as Evs)

Figure 5: misses μ in μm

References: Please italicize species names.

Please decide if in vitro and in vivo is written in italics or not throughout the text.

Authors' Response: Suggestions were fully incorporated into the revised figure.

Reviewer 2 Report

In this manuscript, de Oliveira and colleagues present a timely and well written review on cryptococcal extracellular vesicles (EV). The authors first give an overview of the various types of EVs and the history of main discoveries on cyptococcal EVs. The authors then use this information to explain their various roles in Cryptococcus physiology and virulence. They authors then briefly discuss how EVs can be used for producing vaccines to Cryptococcus. The figures have been well crafted.

While the content of this manuscript is of broad interest to readers of this journal, there is a particular aspect that can be expanded to increase its citation potential. As such, the manuscript should be acceptable for publication after the suggested minor revision.

In lines 291-299, the authors briefly discuss the vaccine potential of cryptococcal EVs. This paragraph merits its very own section, since there is growing interest into how EVs could be exploited therapeutically. In addition to vaccines, the authors should discuss whether targeting proteins in the cryptococcal secretion machinery pathway has or could lead to the development of novel chemotherapies against this family of pathogens.

Author Response

Reviewer 2 Comment: In this manuscript, de Oliveira and colleagues present a timely and well written review on cryptococcal extracellular vesicles (EV). The authors first give an overview of the various types of EVs and the history of main discoveries on cyptococcal EVs. The authors then use this information to explain their various roles in Cryptococcus physiology and virulence. They authors then briefly discuss how EVs can be used for producing vaccines to Cryptococcus. The figures have been well crafted.

While the content of this manuscript is of broad interest to readers of this journal, there is a particular aspect that can be expanded to increase its citation potential. As such, the manuscript should be acceptable for publication after the suggested minor revision.

In lines 291-299, the authors briefly discuss the vaccine potential of cryptococcal EVs. This paragraph merits its very own section, since there is growing interest into how EVs could be exploited therapeutically. In addition to vaccines, the authors should discuss whether targeting proteins in the cryptococcal secretion machinery pathway has or could lead to the development of novel chemotherapies against this family of pathogens.

Authors' Response: Thank you for the encouraging words. As for the application of EVs, we fully agree with the reviewer. Please see the new topic 6 in the revised manuscript, as well as lines 229-234.

Reviewer 3 Report

C. de Oliveira and colleagues have highlighted the most recent findings in the field of extracellular vesicles (EV), particularly in its role for the fungal pathogen Cryptococcus. EVs have been previously understudied and this review discusses the surmounting evidence that will ensure that audiences are familiar with the history of EV in Cryptococcus, familiarize them with the secretory pathways used by this fungi, the virulence factors present in EV, how the host responds to EV as well as the current insights in methodology discussed in this review. 

Although the review is overall well-written, has a good flow and its eventual publication will be impactful to the mycology field and microbial EV community, I felt that some of the illustrations need to be redone/modified prior to being accepted and most notably the authors need some sort of conclusive statement before it can be published. The manuscript just ends abruptly with the discussion of previous and current methodology on generating EV (a challenging task on itself), it needs a closing statement.

For instance, what are some of the most pressing gaps in knowledge in the field that they feel need to be addressed? Do they note any homogeneity in content and composition of EVs at various life-stages of Cryptococcus? What about in different species of Cryptococcus, are most EVs studied derived from C. neoformans? Would we expect differences in virulence factors carried within EV in C. gattii? Would the potential of heterogeneity also be a challenge when generating massive quantities for the use of adjuvants/antigens in vaccine development? 

I am sure the authors have more questions they would like to see answered. 

As far as illustrations, the first figure I am not sure if it would benefit to be converted to a table so that they can further describe the variation in size and its location, or modify it.

Please consider changing the order on the illustration ie. A. Name- ie Apoptotic body (size - I also believe the symbol for μ is missing for Fig 1 and 5), then B. where it's derived from (apoptotic cells) the arrows are not to scale so they are not necessary. Finally, the bilayer membrane in the illustration appears to have the same composition. Do all 3 EVs described have the same lipid membrane composition or different densities? The bilayer should also have some mannoproteins (even GXM) embedded as well, since it was described in line 155 and described in their pre-print [66]. If the authors would like to emphasize the scale (which I believe is important as well) maybe they would benefit to have a cell's organelle/structure as reference for scale?

Minor edits: 

Line 71: 'particles in the 1970s' - insert  'to current times'. 

Line 123-124 needs rephrasing 

Figure 2: Please consider inserting the citation directly to the timeline. Perhaps modify solid line to dotted line  from 1970s to 2000 to show the lapsed time before EV were defined. Also gray bar in timeline is not described in the legend.

Figure 3: The authors would benefit to have Figure 3 present earlier in the manuscript (Biogeneses and secretory pathways...) When they discussed the conventional vs unconventional pathways it was hard to visualize, particularly if you're unfamiliar with the pathway. In the Zoomed EV, I would like to see incorporation of mannoproteins and GXM (as described in Rizzo's 2020 BioRX preprint), in the bilayer either here or on Fig 1. I would also recommend having a black outline for the cell wall layer, the yellow blends into the white background (GRASP and CIN1 are also hard to see, perhaps change color of RNA?)

Line 259-260: Pro-inflammatory and immunoregulatory cytokines are produced in the presence of EV derived from an acapsular and capsular strain. this line will confuse reader, please revise or remove sentence as it's discussed right after. 

Line 338: Discussing ELISAs - what specific target are you using to determine EVs. Also it would be good to indicate the targets for antibodies used to identify EVs.

Figure 4: Alpha in TNF is missing and B for TGF-b. Would the presence of EV, based on cytokine profile, skew macrophages towards a non-protective M2 phenotype?  Out of curiosity, when using sgl1Δ mutant do the authors see a restored function of NO production by macrophages of immunized mice?

Author Response

Reviewer 3 Comment: C. de Oliveira and colleagues have highlighted the most recent findings in the field of extracellular vesicles (EV), particularly in its role for the fungal pathogen Cryptococcus. EVs have been previously understudied and this review discusses the surmounting evidence that will ensure that audiences are familiar with the history of EV in Cryptococcus, familiarize them with the secretory pathways used by this fungi, the virulence factors present in EV, how the host responds to EV as well as the current insights in methodology discussed in this review. 

Although the review is overall well-written, has a good flow and its eventual publication will be impactful to the mycology field and microbial EV community, I felt that some of the illustrations need to be redone/modified prior to being accepted and most notably the authors need some sort of conclusive statement before it can be published. The manuscript just ends abruptly with the discussion of previous and current methodology on generating EV (a challenging task on itself), it needs a closing statement.

For instance, what are some of the most pressing gaps in knowledge in the field that they feel need to be addressed? Do they note any homogeneity in content and composition of EVs at various life-stages of Cryptococcus? What about in different species of Cryptococcus, are most EVs studied derived from C. neoformans? Would we expect differences in virulence factors carried within EV in C. gattii? Would the potential of heterogeneity also be a challenge when generating massive quantities for the use of adjuvants/antigens in vaccine development? 

I am sure the authors have more question they would like to see answered. 

Authors' Response: Thank you for the detailed and constructive analysis. We fully agree with these points and added a final, concluding topic to the revised manuscript. Please see lines 351-370.

Reviewer 3 Comment: As far as illustrations, the first figure I am not sure if it would benefit to be converted to a table so that they can further describe the variation in size and its location, or modify it.

Please consider changing the order on the illustration ie. A. Name- ie Apoptotic body (size - I also believe the symbol for μ is missing for Fig 1 and 5), then B. where it's derived from (apoptotic cells) the arrows are not to scale so they are not necessary. Finally, the bilayer membrane in the illustration appears to have the same composition. Do all 3 EVs described have the same lipid membrane composition or different densities? The bilayer should also have some mannoproteins (even GXM) embedded as well, since it was described in line 155 and described in their pre-print [66]. If the authors would like to emphasize the scale (which I believe is important as well) maybe they would benefit to have a cell's organelle/structure as reference for scale?

Authors' Response: Nice points. We opted for omitting Figure 1, considering the reviewer's comments and the fact that similar illustrations are widely available in the literature.

Reviewer 3 Comment:

Minor edits: 

Line 71: 'particles in the 1970s' - insert  'to current times'. 

Line 123-124 needs rephrasing 

Figure 2: Please consider inserting the citation directly to the timeline. Perhaps modify solid line to dotted line  from 1970s to 2000 to show the lapsed time before EV were defined. Also gray bar in timeline is not described in the legend.

Figure 3: The authors would benefit to have Figure 3 present earlier in the manuscript (Biogeneses and secretory pathways...) When they discussed the conventional vs unconventional pathways it was hard to visualize, particularly if you're unfamiliar with the pathway. In the Zoomed EV, I would like to see incorporation of mannoproteins and GXM (as described in Rizzo's 2020 BioRX preprint), in the bilayer either here or on Fig 1. I would also recommend having a black outline for the cell wall layer, the yellow blends into the white background (GRASP and CIN1 are also hard to see, perhaps change color of RNA?)

Line 259-260: Pro-inflammatory and immunoregulatory cytokines are produced in the presence of EV derived from an acapsular and capsular strain. this line will confuse reader, please revise or remove sentence as it's discussed right after. 

Line 338: Discussing ELISAs - what specific target are you using to determine EVs. Also it would be good to indicate the targets for antibodies used to identify EVs.

Figure 4: Alpha in TNF is missing and B for TGF-b. Would the presence of EV, based on cytokine profile, skew macrophages towards a non-protective M2 phenotype?  Out of curiosity, when using sgl1Δ mutant do the authors see a restored function of NO production by macrophages of immunized mice?

Authors' Response: We took minor edits into fully consideration, as highlighted in the revised manuscript. 

Round 2

Reviewer 2 Report

The manuscript has been improved and is now apt for publication.

Author Response

Thank you for your careful review.

Reviewer 3 Report

I would like to thank the authors for their patience and making modifications to improve the flow and illustrations.

I only have minor comments: 

Please be sure to add back some terminology that was trimmed from the original submission (ex: line 174  multivesicular bodies MVB) 

Images are easier to follow with the manuscript, i would increase font size of the first figure  and last. 

Please revise line 612. Necessarily? 

Figure 3 β and alpha are missing from macrophages cytokine response. 

Author Response

Thank you for the positive comments. Here goes our list of actions during revision:

Reviewer 3 Comment: Please be sure to add back some terminology that was trimmed from the original submission (ex: line 174  multivesicular bodies MVB).

Authors' Response: Thank you for pointing this out. Actually we first mentioned MVBs in line 38: we have described what MVB stands for in this part of the manuscript. 

Reviewer 3 Comment: Images are easier to follow with the manuscript, i would increase font size of the first figure and last. 

Authors' Response: Done. We have increased font size to the maximum allowed to keep figures' properties.

Reviewer 3 Comment: Please revise line 612. Necessarily? 

Authors' Response: We removed the word necessarily. 

Reviewer 3 Comment: Figure 3 β and alpha are missing from macrophages cytokine response. 

Authors' response: We believe this was a problem with pdf conversion, since our original figure contained the appropriate characters. We have corrected this problem.